# Amortized Inference for Heterogeneous Reconstruction in Cryo-EM

**Axel Levy**
Stanford University

**Gordon Wetzstein**
Stanford University

**Julien Martel**
Stanford University

**Frédéric Poitevin**
SLAC National Accelerator Laboratory

**Ellen D. Zhong** *
Princeton University

## Abstract

Cryo-electron microscopy (cryo-EM) is an imaging modality that provides unique insights into the dynamics of proteins and other building blocks of life. The algorithmic challenge of jointly estimating the poses, 3D structure, and conformational heterogeneity of a biomolecule from millions of noisy and randomly oriented 2D projections in a computationally efficient manner, however, remains unsolved. Our method, cryoFIRE, performs *ab initio* heterogeneous reconstruction with unknown poses in an amortized framework, thereby avoiding the computationally expensive step of pose search while enabling the analysis of conformational heterogeneity. Poses and conformation are jointly estimated by an encoder while a physics-based decoder aggregates the images into an implicit neural representation of the conformational space. We show that our method can provide one order of magnitude speedup on datasets containing millions of images without any loss of accuracy. We validate that the joint estimation of poses and conformations can be amortized over the size of the dataset. For the first time, we prove that an amortized method can extract interpretable dynamic information from experimental datasets.

## 1   Introduction

Proteins and other biological macromolecules in the cell function through a finely-tuned choreography of transitions between metastable conformational states. Analyzing the structural heterogeneity of a biomolecule is therefore critical for applications such as drug design and, more generally, for understanding these essential building blocks of life.

In a single particle cryo-electron microscopy (cryo-EM) experiment, an aqueous solution of purified biomolecules is flash-frozen in a thin layer of vitreous ice and imaged with a transmission electron microscope (Fig. 1 (a)). A cryo-EM experiment outputs a large set of unlabeled images, each containing a 2D projection of a unique molecule, whose 3D structure is sampled from some thermodynamic distribution (i.e. a *conformation*) and viewed from an unknown orientation (i.e. a *pose*) $\phi_i = (R_i, \mathbf{t}_i) \in SO(3) \times \mathbb{R}^2$ (Fig. 1 (b)). While *homogeneous* reconstruction methods focus on estimating the average 3D electron scattering potential of the studied molecule (the *consensus* volume), *heterogeneous* methods take into account structural variability and introduce a variable $z_i$ for the conformational state that characterizes the electron scattering potential $\mathcal{V}(., z_i) : \mathbb{R}^3 \to \mathbb{R}$ associated with observation $i$ [5]. Given the image formation model (detailed in Section 3.1), heterogeneous reconstruction can be seen as an inverse problem where $\mathcal{V}$, $z_i$ and $\phi_i$ must be inferred from the observed images $Y_i$. A graphical formulation of the problem is given in Fig. 1 (c).

---

*Correspondence to: zhonge@princeton.edu

36th Conference on Neural Information Processing Systems (NeurIPS 2022).

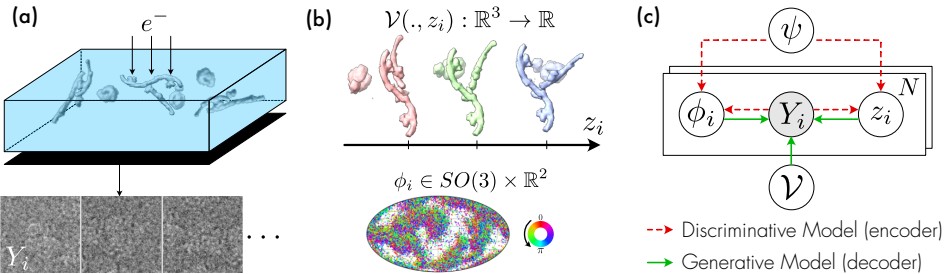

Figure 1: (a) Illustration of a cryo-EM experiment. Molecules are frozen in a random orientation $\phi_i$ and in a random conformation $z_i$. Their electron scattering potential $\mathcal{V}(., z_i)$ interacts with an electron beam resulting in noisy projected images $Y_i$ on the detector. (b) (Top) Example scattering potentials visualized as an isosurface. (Bottom) Poses are characterized by a rotation in $SO(3)$ and a translation in $\mathbb{R}^2$ (not shown). (c) Graphical model for our method. The encoder, parameterized by $\psi$, is a discriminative model that predicts $\phi_i$ and $z_i$ from $Y_i$ while the decoder is a generative model that outputs a noise-free estimation of the image.

Driven by recent advances in data collection capabilities, the number of images collected per cryo-EM experiment has been steadily increasing, now reaching millions to tens of millions [1, 19]. Established methods for heterogeneity analysis break down 3D reconstruction into two alternating iterative refinement steps: 1) the *latent* pose $\phi_i$ and conformational states $z_i$ are first estimated from the images and the current estimate of $\mathcal{V}$, and 2) the estimation of $\mathcal{V}$ is then updated using the current estimates of the latent variables. The primary computational bottleneck of this approach is pose estimation, which is done by rendering images from view points distributed on the the 5-dimensional space $SO(3) \times \mathbb{R}^2$. Although previous methods came up with accelerated branch-and-bound strategies [14, 24, 38], state-of-the-art methods for pose search raise two problems: (1) solving this problem for a single image is time consuming, especially when the images are rendered using a neural model [39] and (2) the pose needs to be solved independently for each image in the dataset and does not use the fact that similar images are likely associated with the similar variables. Furthermore, most heterogeneous reconstruction algorithms limit their applicability by relying on an upstream homogeneous reconstruction, treating poses as known to simplify the optimization problem for $z_i$ or due to the sheer computational cost of pose search.

In this work, we focus on *ab initio* heterogeneous reconstruction, meaning that reconstruction occurs *de novo* without any upstream estimation of the latent pose or conformation variables. We present a method that amortizes (over the size of the dataset) the cost of inference by learning a function that directly maps observed images to estimates of the latent variables. Our method, cryoFIRE (Fast heterogeneous ab Initio Reconstruction for cryo-EM), leverages an amortized approach to address the increasing size of datasets, while tackling the opportunity to learn complex conformational spaces of dynamic proteins. Our contributions include:

- An autoencoder-based architecture with a tailored loss function that processes images 10 times faster than existing methods and enables amortization of the runtime over the size of the dataset,

- A demonstration that our model accurately learns the structure of a low-dimensional manifold in the conformational space on benchmark synthetic datasets with both discrete and continuous heterogeneity, and

- To our knowledge, the first instance of amortized inference for *ab initio* heterogeneous reconstruction of experimental cryo-EM datasets.

## 2   Related Work

Heterogeneous reconstruction methods in cryo-EM can be differentiated according to the way the latent pose $\phi_i$ and conformation state $z_i$ are estimated [5]. The vast majority of approaches assume that $\phi_i$ are previously estimated. We describe related work in terms of how $\phi_i$ is estimated, how $z_i$ is

estimated (without pose inference), and progress towards joint amortization of all latent, unknown variables in cryo-EM.

**Expectation-Maximization over Conformations** With an Expectation-Maximization (EM) algorithm, posterior distributions over the conformational states are computed (or approximated) for each image at expectation step. In cryo-EM, this method was first popularized by RELION with 3D Classification [29] which implemented a discrete class indicator $z_i \in \{1, \ldots, K\}$, where $K$ is the number of classes defined by the user. More recently, a continuous reconstruction method called *multi-body refinement* was made available in RELION [18]. In this approach, a homogeneous reconstruction is segmented by the user into a set of rigid bodies, where each rigid body is free to move relative to the others. The relative pose (orientation and translation) of each body is subsequently estimated through EM for each particle, and further reduced through Principal Component Analysis (PCA). Various methods have been developed to estimate the conformational space as a linear deformation around a known reference. HEMNMA [9] uses the normal modes, or eigenvolumes, of the reference to model its deformation and proceeds through a projection-matching-based elastic alignment of each single-particle image with the reference structure to yield a reduced representation of the variability in the dataset. 3DVA [23] relaxes the dependence on known normal modes and instead implements a variant of the EM algorithm for Probabilistic PCA which iteratively updates the eigenvolumes and the projection of the particles on them, until convergence, following [32]. Other approaches explicitly estimate the deformation field mapping the reference structure to the best-fit structure for each image. In Herreros et al. [8], the deformation field is defined as a linear expansion over Zernike polynomials, with $z_i$ as their coefficients. In 3DFlex [22], the deformation field is generated from $z_i$ through a neural flow generator, alleviating the linear constraint. Formally, each $z_i$ can be seen as a point estimate that maximizes the posterior distribution over conformational states. An in-depth review of EM-based reconstruction method can be found in [31]. Other computational approaches like Markov Chain Monte Carlo have been explored for *ab initio* heterogeneous reconstruction in [15], introducing a mathematical framework for representing deformable molecules ("hyper-molecules"), but remain a prototype implementation.

**Amortization over Conformations** Instead of optimizing each variable independently, amortized inference learns a function $q_\psi$, parameterized by $\psi$, that maps images $Y_i$ to probability (posterior) distributions over the space of latent variables [11]. If $N$ represents the size of the dataset, amortized inference therefore replaces the estimation of $N$ latent variables with the learning of $\psi$, which complexity (number of dimensions) does not scale with $N$. Previous methods explored the possibility of using amortized inference to estimate the conformational state $z_i$, in a setting where the poses were known. CryoDRGN [37] introduced amortized variational inference to estimate the conformational state $z_i$ in the setting where poses are known. In cryoDRGN [37], distributions over the conformational states are predicted by the encoder of a Variational Autoencoder (VAE) [11, 12] and the conformational space is parameterized with an implicit neural representation. E2GMM [4] and cryoFold [40] both reconstruct a deformable atomic representation of the molecule and train an encoder to associate each image with a low-dimensional $z_i$ that characterizes the deformation. These two methods require the initialization of the backbone structure of the molecule. Although these methods can be used to analyze the structural heterogeneity in a dataset, they assume the poses to be given by another reconstruction method. In cryoDRGN-BNB [38] and cryoDRGN2 [39] the poses are not given anymore but instead are estimated using an exhaustive search strategy. The estimation of the poses therefore does not amortize over the size of the dataset and, in spite of a branch-and-bound approach [14], the 5D pose search remains the most computationally expensive step in the pipeline.

**Amortization over Poses** In the context of homogeneous reconstruction, previous methods used amortized inference to predict the latent variables. In cryoVAEGAN [17], Miolane *et al.* showed that the in-plane rotation and the contrast transfer function (CTF) parameters could be jointly estimated in the latent space of an encoder. CryoGAN [6] showed that homogeneous reconstruction could be achieved with a generative framework using a discriminative loss to avoid explicitly recovering the poses, considered as "nuisance" variables. SpatialVAE [2] showed that the translations and the in-plane rotations in 2D images could be estimated with a VAE-based architecture, and was later generalized to other transformations in [34]. CryoPoseNet [20] first demonstrated the possibility of using an autoencoder architecture with synthetic datasets. CryoAI [16] later introduced the *symmetric loss* to help the model avoid local minima and showed that homogeneous reconstruction could be done on experimental datasets. All of these methods, however, reconstruct a 3D *consensus* volume and do not address the question of conformational heterogeneity.

**Joint Amortization** Rosenbaum *et al.* [26] demonstrated heterogeneous reconstruction from unknown poses in a jointly amortized framework on simulated data and with strong priors on the structure given by an initial atomic model. To the best of our knowledge, no amortized inference technique has been shown in a more general setting on *experimental datasets* with unknown poses. Here, cryoFIRE uses a implicit neural representation for the conformational space that can reconstruct experimental datasets with accuracy comparable to the state of the art at a fraction of the compute time.

## 3 Methods

### 3.1 Image Formation Model

In single particle cryo-EM, probing electrons interact with the electrostatic potential created by the molecules embedded in a thin layer of vitreous ice (Figure 1 (a)). During reconstruction, we assume that this potential can be broken down into spatially bounded independent potentials (*volumes*) created by individual molecules. Each volume $V_i$ can be seen as a mapping from $\mathbb{R}^3$ to $\mathbb{R}$ and is indexed by $i \in \{1, \ldots, N\}$. We assume that the volumes $\{V_i\}_{i=1,\ldots,N}$ are drawn independently from a probability distribution $\mathbb{P}_V$ supported on a low-dimensional manifold (the *conformational space*) in the space $\mathcal{F}(\mathbb{R}^3, \mathbb{R})$ of all possible 3D potentials ($\mathcal{F}(A, B)$ is the set of functions from $A$ to $B$). More specifically, we assume there exist $d \in \mathbb{N}$ and $\mathcal{V} : \mathbb{R}^3 \times \mathbb{R}^d \to \mathbb{R}$ such that $\mathbb{P}_V$ is supported on the conformational space $\{\mathcal{V}(., z), z \in \mathbb{R}^d\}$.

In the sample, each molecule is in an unknown orientation $R_i \in SO(3) \subset \mathbb{R}^{3 \times 3}$ in the frame of the observer. The probing electron beam interacts with the electrostatic potential and its projections,

$$Q_i : (x, y) \mapsto \int_t \mathcal{V}\left(R_i \cdot [x, y, t]^T, z_i\right) dt \tag{1}$$

are considered mappings from $\mathbb{R}^2$ to $\mathbb{R}$. The interaction between the beam and the lens is modeled by the Point Spread Function (PSF) $P_i$. Imperfect centering of the molecule in the image is characterized by small translations $\mathbf{t}_i \in \mathbb{R}^2$. Finally, taking into account signal arising from the vitreous ice into which the molecules are embedded as well as the non-idealities of the lens and the detector, each image $Y_i$ is generally modeled as

$$Y_i = T_{\mathbf{t}_i} * P_i * Q_i + \eta_i \tag{2}$$

where $*$ is the convolution operator, $T_{\mathbf{t}}$ the $\mathbf{t}$-translation kernel and $\eta_i$ white Gaussian noise on $\mathbb{R}^2$ [35, 28].

The Fourier-Slice Theorem [3] (FST) avoids the computation of integrals and convolutions in Eq. (2) and is also satisfied with the Hartley transform:

$$\mathcal{H}_{2D}[Q_i] = \mathcal{S}_i\left[\mathcal{H}_{3D}[\mathcal{V}(., z_i)]\right], \tag{3}$$

where $\mathcal{H}_{2D}$ and $\mathcal{H}_{3D}$ are the 2D and 3D Hartley transform operators [7] (real minus imaginary parts of the Fourier transform). The "slice" operator $\mathcal{S}_i$ is defined such that for any $\hat{V} : \mathbb{R}^3 \to \mathbb{R}$,

$$\mathcal{S}_i[\hat{V}] : (k_x, k_y) \mapsto \hat{V}\left(R_i \cdot [k_x, k_y, 0]^T\right). \tag{4}$$

Finally, if $\hat{Y}_i = \mathcal{H}_{2D}[Y_i]$ and $\hat{\mathcal{V}}(., z_i) = \mathcal{H}_{3D}[\mathcal{V}(., z_i)]$, the image formation model in Hartley space can be expressed as

$$\hat{Y}_i = \hat{T}_{\mathbf{t}_i} \odot C_i \odot \mathcal{S}_i[\hat{\mathcal{V}}(., z_i)] + \hat{\eta}_i, \tag{5}$$

where $\odot$ indicates element-wise multiplication, $C_i = \mathcal{H}_{2D}[P_i]$ is the Contrast Transfer Function (CTF), $\hat{T}_{\mathbf{t}}$ the $\mathbf{t}$-translation operator or phase-shift in Fourier space and $\hat{\eta}_i$ represents complex white Gaussian noise on $\mathbb{R}^2$. See Supplement E for a discussion on the discretization step.

### 3.2 Overview of cryoFIRE

Fig. 2 summarizes the architecture of cryoFIRE. Images $Y_i$ are fed into an encoder, parameterized by $\psi$, that predicts a pose $\phi = (R_i, \mathbf{t}_i)$ and a conformational state $z_i$. $R_i$ is used to rotate a grid of $D^2$ 3D-coordinates in Hartley space. These coordinates are concatenated with the conformational state before being fed into a neural representation of the function $\hat{\mathcal{V}}_\theta$ (in Hartley space), parameterized by $\theta$.

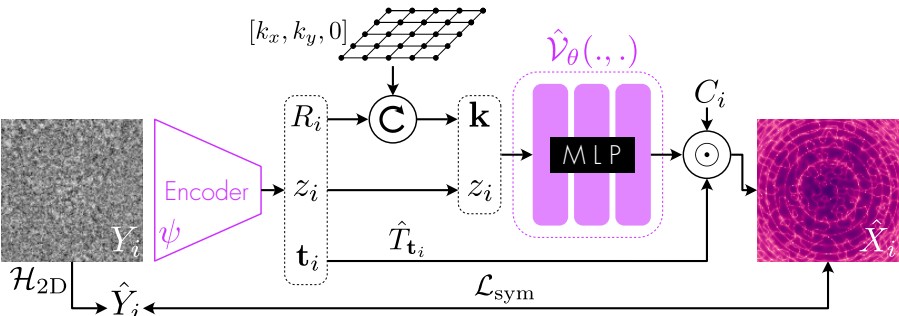

Figure 2: Architecture of cryoFIRE. An encoder predicts the latent variable $z_i$ (conformational state), $R_i$ (rotation) and $\mathbf{t}_i$ (translation) for each image. These variables are used by a physics-based decoder that reproduces the image formation model (in Hartley space) and contains a neural representation $\hat{\mathcal{V}}_\theta$ of the conformational space. Reconstructed and measured images are compared with the symmetric loss.

$\hat{\mathcal{V}}_\theta$ is a mapping from $\mathbb{R}^{3+d}$ to $\mathbb{R}$ and can be seen as a parametric representation of the conformational space in Hartley domain (a manifold of dimension $d$). Queried for all $\mathbf{k}$ in the grid, the neural representation outputs a set of real values corresponding to a discrete sampling of the slice defined in Eq. (4). Based on the estimated translation $\mathbf{t}_i$ and given CTF parameters $C_i$, the rest of the image formation model described in Eq. (5) is simulated to obtain $\hat{X}_i$, a noise-free estimation of $\hat{Y}_i$. The whole forward model is refered to as $\Gamma_{\psi,\theta}$, such that $\hat{X}_i = \Gamma_{\psi,\theta}(Y_i)$. Pairs of measured and reconstructed images are compared using the symmetric loss (see Section 3.5) and gradients are backpropagated throughout the differentiable model in order to optimize both the encoder and the neural representation.

### 3.3 Discriminative Model

In cryoFIRE, an encoder acts as a discriminative model by mapping images to estimates of $\phi_i$ and $z_i$. The encoder is structured sequentially with the following components:

1. A Convolutional Neural Network (CNN) containing 7 convolutional layers extracts high-level visual features from images and divides the width and height of images by 32. The architecture of the CNN is inspired from the first layers of VGG16 [30], known to perform well on visual tasks.

2. A *shared* Multi-Layer Perceptron (MLP) with 2 hidden layers that outputs a feature $y_i$ of dimension 256.

3. A *conformation* MLP that maps $y_i$ to $z_i$.

4. *Rotation* and *translation* MLPs that map the concatenation of $y_i$ and $z_i$ respectively to $R_i$ and $\mathbf{t}_i$. The pose prediction is conditioned on $z_i$ since rotations and translations are only defined for a given conformation of the molecule and $z_i$ contains, with a small number of dimensions, all the required information to determine the conformational state. The rotation is represented in the 6-dimensional space $\mathcal{S}^2 \times \mathcal{S}^2$ [41] as it was shown to lead to the best results for the rotation prediction in [20].

We experimented with both variational (predicting an approximate posterior distribution over latent variables) and non-variational (predicting a point estimate) approaches and did not observe any significant performance difference (although the variational approach is more computationally expensive). Full details about the architecture of the encoder are given in Supplement A.

### 3.4 Generative Model

The generative model is a simulation of the image formation model in Hartley space, in order to make use of the FST (Section 3.1). The forward pass is differentiable with respect to $z_i$, $R_i$ and $\mathbf{t}_i$. For each $i$, a grid of $D^2$ coordinates on the x-y plane in Hartley space are rotated by $R_i$ (mapped from $\mathcal{S}^2 \times \mathcal{S}^2$ to $\mathbb{R}^{3\times3}$ using Gram–Schmidt orthogonalization) *via* a matrix multiplication. Rotated

points $\mathbf{k}$ are positionally encoded with a bank of $D/2$ random frequencies [33] and independently concatenated with the conformational state $z_i$ before being fed into an MLP mapping $\mathbb{R}^{3+d}$ to $\mathbb{R}$. This neural network is an implicit representation of the function $\hat{\mathcal{V}}$, where $\hat{\mathcal{V}}(.,z) = \mathcal{H}_{3D}[\mathcal{V}(.,z)]$. Additional implementation details about the neural network are given in Supplement B.

The CTF is defined by the *defocus* parameters and the *astigmatism angle* (provided by the simulator or by an external software like CTFFIND [25] for experimental datasets). It is then applied to the slice of size $D^2$ outputted by the neural representation. Finally, the slice is element-wise multiplied by $\hat{T}_{\mathbf{t}_i}$, the Hartley transform of the $\mathbf{t}_i$-translation kernel.

### 3.5 Training Procedure

In the setting where poses are unknown, one cannot distinguish, given a set of 2D projections, two molecules with different handedness. This is called the "handedness ambiguity" [27]. Ref. [16] showed that this leads amortized inference techniques to get stuck in local minima where the predicted molecule contains spurious planar symmetries. A solution suggested to alleviate this problem is to use the *symmetric loss*

$$\mathcal{L}_{\text{sym}} = \sum_{i \in \mathcal{B}} \min \left\{ \|\hat{Y}_i - \Gamma_{\psi,\theta}(Y_i)\|_2^2, \|\mathcal{R}_\pi[\hat{Y}_i] - \Gamma_{\psi,\theta}(\mathcal{R}_\pi[Y_i])\|_2^2 \right\}, \qquad (6)$$

where $\mathcal{B}$ is a batch of indices and $\mathcal{R}_\pi$ applies an in-plane rotation of $\pi$ on images (we refer the reader to [16] for an ablation study on the symmetric loss and its instrumental role in the amortized inference of poses). In order to enable the model to converge to a consensus volume, we disable the optimization of the conformation MLP at the start of the training. During this "pose-only phase", $z_i$ are randomly sampled from a normalized Gaussian distribution.

## 4 Results

We evaluate cryoFIRE for *ab initio* heterogeneous reconstruction and compare it with the state-of-the-art method cryoDRGN2 [39]. We first validate that using an encoder to predict poses, instead of performing an exhaustive pose search, enables us to reduce the runtime of heterogeneous reconstruction on a synthetic dataset. We show that the encoder is able to accurately predict $\phi_i$ and $z_i$ for images it has never processed during training, thereby validating the ability of an encoder-like architecture to amortize the runtime over the size of the dataset. Secondly, we show that cryoFIRE enables the detection of discrete heterogeneity in a dataset, by clustering images in the $z_i$-space. Finally, we show that our method can perform heterogeneous reconstruction of real data, a first for a method that jointly amortizes the estimation of poses and conformations.

### 4.1 Runtime Improvement and Amortization

**Experimental Setup** We prepare three synthetic datasets of varying sizes – *small* (50k images), *medium* (500k images) and *large* (5M images) (Supplement C). Images are generated using a simulation of the image formation model on ground truth volumes of the 80S ribosome. Each dataset contains a mix of projections with $90\%$ sampled from the one of the volumes (*major*) and the rest from the other volume (*minor*). CTFs are drawn randomly with a log-normal distribution over the defocus range. We add Gaussian noise with variance $\sigma^2 = 10$ to the normalized projections (SNR=$-10$dB). With cryoFIRE, we fix $d = 8$ and activate the conformation MLP after the model has seen 1.5M images. We compare our method with cryoDRGN2 [39] with default parameters. Pixels' intensities are set to 0 outside of a circle of radius 32 pixels for pose search. With cryoDRGN2, pose search is done every 5 epochs for the small datasets and every epoch for the medium and large datasets. We train the models on a single NVIDIA A100 SXM4 40GB GPU. Images of size $D = 128$ are fed by batches of maximum sizes (128 for cryoFIRE, 32 for cryoDRGN2) and loaded on the fly using multi-threading on 16 CPUs. The model is optimized with the ADAM optimizer [10] and a learning rate of $10^{-4}$. Convergence is assessed via visual inspection of the pose prediction accuracy. Once convergence is reached on the *train set*, latent variables can be estimated on a *test set* of 10k images in order to validate that the encoder is not just memorizing the training set. With cryoDRGN2, poses from the test set are estimated with a randomly-initialized pose search on the model obtained at train time.

Table 1: Heterogeneous reconstruction on datasets of varying sizes. Each dataset contains a mix of images, $90\%$ from the *major* volume and $10\%$ from the *minor* volume. After convergence on a train set, each method estimates the latent variables $(R_i, \mathbf{t}_i, z_i)$ on a test set. We report the resolution (Res. in pixels, $\downarrow$), the confusion error ($\downarrow$), the rotation accuracy (Rot. in degrees, $\downarrow$) and the translation accuracy (Trans. in pixels, $\downarrow$). Convergence is detected on the pose accuracy (ep. = epochs).

| Dataset / Method | Time | Confusion | Res. (major/minor) | Rot. (Med/MSE) | Trans. (Med/MSE) |
|---|---|---|---|---|---|
| *Small* (Train: 50k / Test: 10k) | | | | | |
| cryoDRGN2 (train) | **1:21h** (20 ep.) | **0.00005** | **2.4 / 2.8** | **0.8 / 0.8** | **0.007 / 0.01** |
| **cryoFIRE** (train) | 1:33h (70 ep.) | 0.0004 | 2.6 / 3.2 | 2.3 / 2.6 | 0.09 / 0.1 |
| cryoDRGN2 (test) | 3 min. (1 ep.) | **0** | — | **0.8 / 0.8** | **0.006 / 0.01** |
| **cryoFIRE** (test) | **11 sec.** (1 ep.) | 0.001 | — | 2.6 / 2.7 | 0.2 / 0.3 |
| *Medium* (Train: 500k / Test: 10k) | | | | | |
| cryoDRGN2 (train) | 5:10h (2 ep.) | 0.002 | **2.5 / 3.0** | **0.8 / 0.9** | **0.007 / 0.01** |
| **cryoFIRE** (train) | **1:28h** (7 ep.) | **0.0008** | 2.7 / 3.2 | 2.7 / 2.9 | 0.1 / 0.2 |
| cryoDRGN2 (test) | 3 min. (1 ep.) | **0.0001** | — | **0.8 / 0.9** | **0.007 / 0.01** |
| **cryoFIRE** (test) | **11 sec.** (1 ep.) | **0.0001** | — | 2.3 / 2.5 | 0.1 / 0.2 |
| *Large* (Train: 5M / Test: 10k) | | | | | |
| cryoDRGN2 (train) | 21:37h (1 ep.) | 0.002 | **2.3 / 2.6** | **0.8 / 1.6** | **0.01** / 1.2 |
| **cryoFIRE** (train) | **1:55h** (1 ep.) | **0.0002** | **2.3** / 2.7 | 1.5 / 1.7 | 0.1 / **1.0** |
| cryoDRGN2 (test) | 3 min. (1 ep.) | **0** | — | **1.0 / 1.0** | **0.007 / 0.1** |
| **cryoFIRE** (test) | **11 sec.** (1 ep.) | **0** | — | 1.2 / 1.4 | 0.1 / 0.2 |

**Metrics** We compute the median and the mean of the angular error on the view direction, in degrees, and the median and mean square error of the predicted $\mathbf{t}_i$, in pixels. Before computing pose errors, a rigid 6D-body alignment is applied on the set of predicted poses to align the model into the same global reference frame as the ground truth volume. We perform a Principal Component Analysis (PCA) on the set of predicted $z_i$'s and, looking at the first principal component, we group images in two classes, according to the cluster they fall into. The confusion error is defined as the ratio of misclassified images over the total number of images. After convergence, we generate two volumes by taking the centroids of the clusters in $z_i$ space and using them as inputs for the neural network $\hat{\mathcal{V}}_\theta$ on them. We report the resolution (in pixels) obtained on the major ($90\%$) and on the minor ($10\%$) volumes. The reported resolution is the inverse of the maximum frequency for which the Fourier Shell Correlation is above $0.5$.

**Results** We report the runtime and quantitative metrics in Table 1. At train time, the average time for processing one image is 19 ms ($5{:}10\text{h} / (2 \times 500\,000)$) with cryoDRGN2 vs. 1.5 ms ($1{:}28\text{h} / (7 \times 500\,000)$) for cryoFIRE. (The runtime of cryoDRGN2 is lower with the small dataset because pose search is only done every 5 epochs). At test time, only the encoder of cryoFIRE is used, which further decreases the runtime per image (1.1 ms). The accuracy obtained at test time on $\phi_i$ and $z_i$ validates that our method effectively amortizes the estimation of latent variables, and the errors on the test sets decrease with the size of the dataset provided for training. The accuracy of predicted poses is slightly worse with cryoFIRE than with an exhaustive pose search strategy (see Supplement E for a comparison with a fully non-amortized method [24]). The quantitative resolution of the reconstructed volumes remains however very similar with both methods (see Supplement E for a qualitative comparison).

## 4.2 Heterogeneous Reconstruction of Synthetic Datasets

**Experimental Setup** We prepare three heterogeneous datasets of 100k images of the *Plasmodium falciparum* 80S ribosome (Supplement C). The three datasets contain an equal mix of projections from $K \in \{2, 3, 10\}$ different ground truth volumes, generated sequentially along a reaction path that corresponds to a rotation of the small ribosomal subunit relative to the large ribosomal subunit identified in [37]. The image formation model is simulated with the same parameters as in Section 4.1. We run cryoFIRE with $d = 8$ and perform a PCA on the set of predicted $z_i$'s after 95 epochs.

**Results** In Fig. 3, we plot the first principal component (PC1) of the set of $z_i$'s vs. the ground truth index of the volume each image was generated from. With the dataset containing $K$ conformations, images can be clustered in $K$ groups, based on the value of PC1 ($K \in \{2, 3\}$). Looking for the

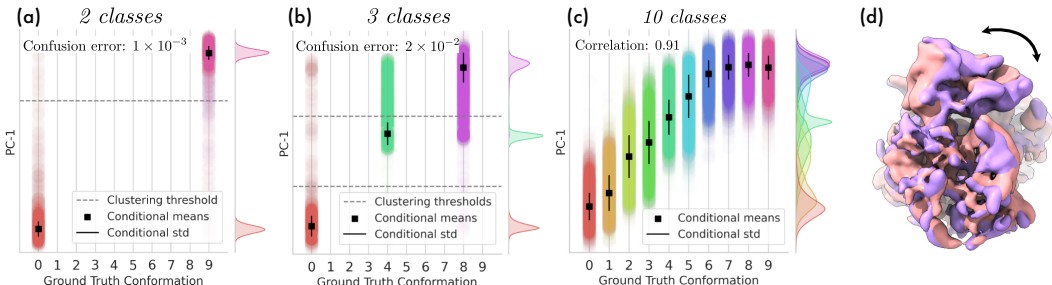

Figure 3: Conformational heterogeneity analysis on synthetic datasets. On (a-b-c), the y-axis shows first principal component (PC1) among the $z_i$'s. We indicate means $\pm$ standard deviations of the PC1s, conditioned on the ground truth conformation. On (a-b) dashed lines are the threshold used to compute the confusion error. Spearman correlation [13] is indicated in (c). (d) is a superposition of two volumes obtained by sampling two points along the PC1 axis for the experiment *2 classes*.

optimal thresholds on PC1, we compute a confusion error of $1 \times 10^{-3}$ (resp. $2 \times 10^{-2}$) on the *2 classes* (resp. *3 classes*) dataset. On the *10 classes* dataset, the 10 conformations cannot be separated but the value of PC1 embeds information about the movement of the ribosome, as indicated by the Spearman correlation [13] between the index of the ground truth conformation and PC1. Sampling different values for PC1 (and setting the other components to zero), we can use the neural representation $\hat{\mathcal{V}}_\theta$ to generate a set of volumes from a set of conformational states $z_i$. Qualitative results are given in Fig. 3 (d), showing an accurate reconstruction of the movement of a subunit of the ribosome.

### 4.3 Heterogeneous Reconstruction of an Experimental Dataset

**Experimental Setup** We use the publicly available dataset EMPIAR-10180 [21] of a pre-catalytic spliceosome (Supplement C). We run cryoFIRE with $d = 8$, activate the conformation MLP after 30 epochs and train for a total of 100 epochs. Results from cryoDRGN2 [39] and cryoDRGN-BNB [38] on the same dataset are obtained from [39].

**Results** The first two components of a PCA on the set of predicted $z_i$'s is shown in Fig.4. By traversing the conformational space along the direction of PC1 and generating volumes on set of 5 points, cryoFIRE generates a trajectory, showing a large flexing motion of the spliceosome. CryoFIRE qualitatively recovers the non-uniform distribution of viewing directions, avoids the local minima cryoDRGN-BNB falls into, and reconstructs volumes which qualitatively match the state of the art. See Supplement D for more results and a reconstruction on an experimental dataset of the 80S ribosome (EMPIAR-10028 [36]).

## 5 Discussion

As the resolving power of a sampling method scales with the size of the dataset, an increasing number of images needs to be collected in cryo-EM in order to extract meaningful structural information. CryoFIRE answers the need for reconstruction methods that scale appropriately with the dataset size: heterogeneous *ab initio* reconstruction on a dataset of 5M images can be performed within 2 hours. CryoFIRE is also the first reconstruction method to perform joint amortization of poses and conformations on an experimental cryo-EM dataset, opening the door to fast analysis of structural heterogeneity on real datasets.

The interpretability of the conformational space remains an open question. Since distances in this space are not meaningful ($z_i$ conditions the volume *via* a *nonlinear* decoder), the probability distribution cannot be straightforwardly interpreted in that space. In the case of discrete heterogeneity, general quantities like the number of states or the relative populations of the states can be reliably inferred from the conformational space by clustering the predicted conformational states. In the case of continuous heterogeneity, dimensionality reduction methods like PCA can highlight directions of maximal variance and, by visual inspection of the reconstructed volumes, cryoFIRE can help understanding the main degrees of freedom of a continuously deformable molecule. However, "small" and "large" deformations can stem from similar changes in the conformational space and no physically

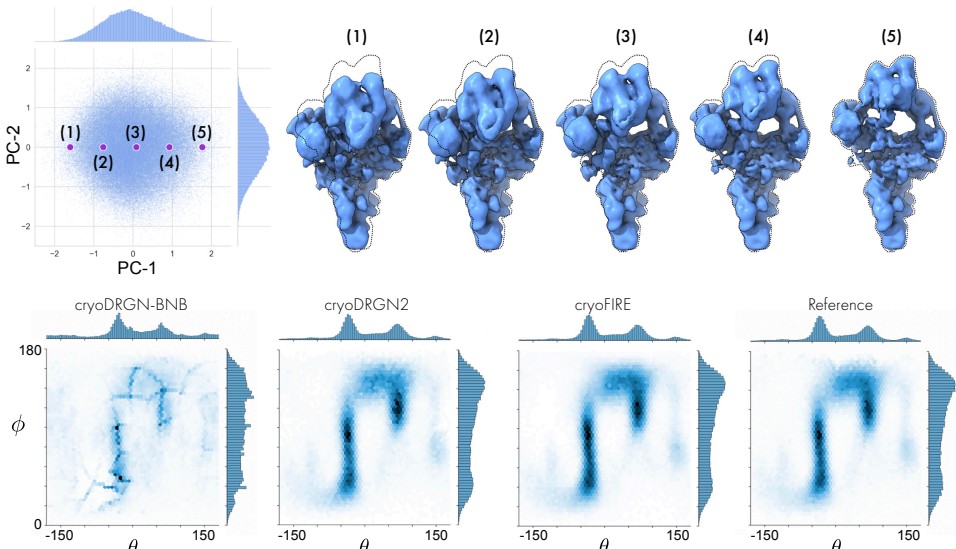

Figure 4: *Ab initio* heterogeneous reconstruction of an experimental dataset of the pre-catalytic spliceosome (EMPIAR-10180 [21]). (Top left) Distribution of the two first principal components of $z_i$ and sampling points along the first component. (Top right) Structures generated by traversing along PC1. The outline indicates the edges of volume (5). (Bottom) Distribution of the viewing directions. The "reference" shows the viewing directions published in [21].

interpretable notion of distance is currently associated with the conformational space: providing this space with a physically interpretable metric is an interesting avenue for future work.

It is worth highlighting that the parameterization proposed here is not uniquely defined: a rotation of the molecule can equivalently be represented by $R_i$ or by a change of conformation state $z_i$. Empirically, cryoFIRE decorrelates poses and conformation by relying on a "pose-only phase" at the beginning of training, but nothing explicitly prevents $z_i$ from containing information about $R_i$. Future work could explore improvements in the optimization procedure to further enforce minimal mutual information between poses and conformation.

Finally, we note that the accuracy of pose estimation (especially on translations) is lower with cryoFIRE than with exhaustive pose search methods. Inaccurate translation prediction can lead to "blurry" reconstructions and therefore limits the resolution of reconstructed volumes. This is a direct reflection of the fact that amortization maximizes a non-tight lower-bound of the likelihood, allowing for faster inference at the cost of accuracy [5]. Future work could investigate a hybrid approach that would estimate poses with an encoder at the beginning of training and switch, at the end, to a local pose search initialized from encoder-estimated poses.

**Broader Impacts Statement** The cryo-EM field is generating data at a rapidly increasing pace, potentially leading to wasteful storage and inefficient computing. The design of more efficient methods such as CryoFIRE provides a path to mitigate sustainability risks. In order to better compare modern reconstruction methods, careful attention must be paid toward designing common datasets and evaluation metrics. By providing an open-source implementation of CryoFIRE upon publication, together with benchmark metrics, we hope to make cryo-EM research accessible to a broader class of researchers. We also acknowledge that, by pushing further the boundaries of biological research, our method may facilitate the development of harmful biologics, however this is outweighed by its significant societal benefits. We encourage the cryo-EM community to play wisely with cryoFIRE.

# 6 Acknowledgements

This work was supported by the U.S. Department of Energy, under DOE Contract No. DE-AC02-76SF00515 and the SLAC LDRD program. We acknowledge the use of the computational resources at the SLAC Shared Scientific Data Facility (SDF) and Princeton Research Computing.

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
