# Supplementary Material
# Amortized Inference for Heterogeneous Reconstruction in Cryo-EM

**Axel Levy**◉
Stanford University

**Gordon Wetzstein**◉
Stanford University

**Julien Martel**◉
Stanford University

**Frédéric Poitevin**◉
SLAC National Accelerator Laboratory

**Ellen D. Zhong**◉
Princeton University

## A  Architecture of the Encoder

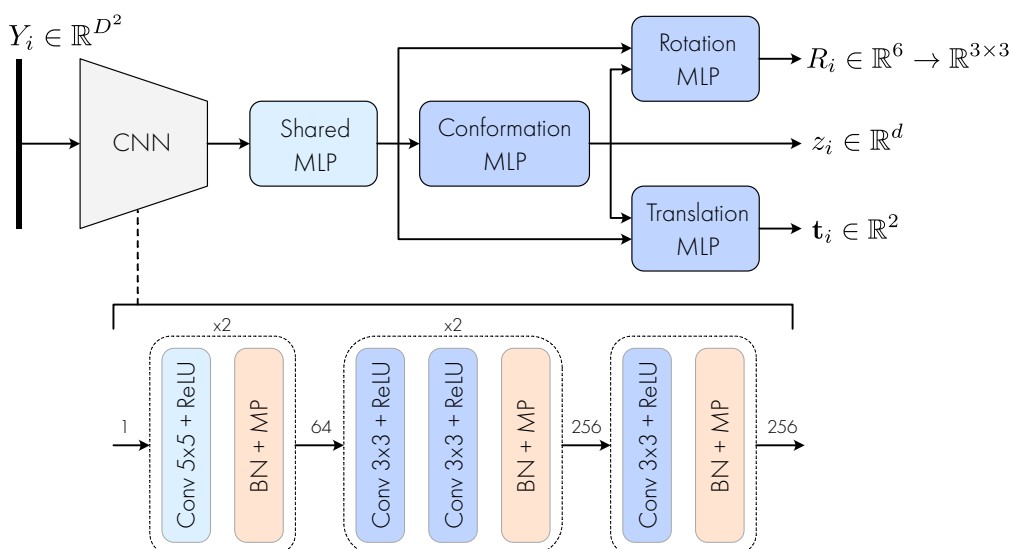

Figure S1: Architecture of the encoder. The image first passes through a CNN (BN = Batch Normalization ; MP = 2-by-2 Max-Pooling). The numbers above the arrows indicate the number of channels.

Fig.S1 shows the architecture of the encoder. The image is fed into a Convolutional Neural Network (CNN) that produces 256 channels of sizes $(D/32)^2$, where $D$ is the resolution of the images in pixels. The output is flattened and fed through a *shared* Multi-Layer Perception (MLP) with 4 hidden layers of dimension 256. The output $y_i \in \mathbb{R}^{256}$ is fed through the *conformation* MLP that contains 3 hidden layers of dimension 256 and one output layer of dimension $d$, interpreted as $z_i$. Concatenated vectors $(y_i, z_i)$ pass through the *rotation* and the *translation* MLPs to produce, respectively, $R_i$ and $\mathbf{t}_i$.

36th Conference on Neural Information Processing Systems (NeurIPS 2022).

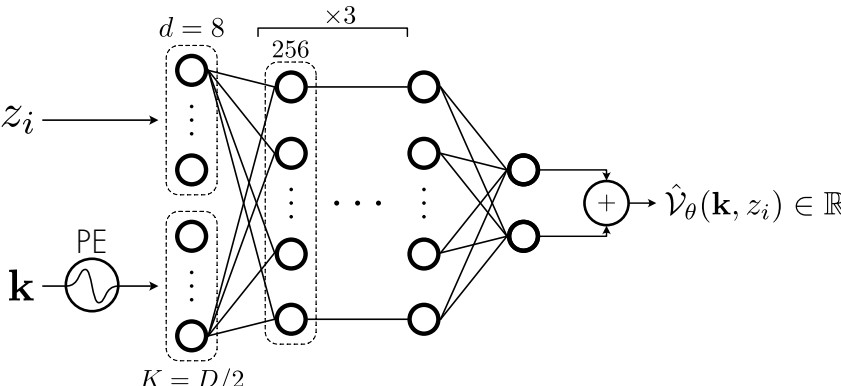

Figure S2: Architecture of the neural representation $\hat{\mathcal{V}}_\theta : \mathbb{R}^{3+d} \rightarrow \mathbb{R}$, approximation of the Hartley transform of $\mathcal{V}$. $\mathbf{k}$ is positionally encoded (PE) with $K = D/2$ frequencies.

## B   Neural Representation

The architecture of the neural representation $\hat{\mathcal{V}}_\theta$ is summarized in Fig. S2. The coordinates $\mathbf{k} \in \mathbb{R}^3$ are positionally encoded with $(D/2)$ randomly sampled frequencies (with Gaussian distribution) [4]. Coordinates of image pixels are defined on a lattice spanning $[-1, 1]^2$. We then restrict evaluation of the model to the pixels in a circle of radius 1, thus the support of our model is defined on a sphere with radius 1. The network contains 4 hidden layers of dimensions 256 and outputs a vector of dimension 2, interpreted as the Cartesian representation of the (complex) number $\mathcal{F}_{3D}[\mathcal{V}](\mathbf{k}, z_i)$ ($\mathcal{F}_{3D}$ is the 3D Fourier transform on spatial coordinates). The Hartley transform is obtained by subtracting the imaginary part from the real part of the Fourier transform. The advantage of working in Fourier space is that we can make use of the fact

$$\mathcal{F}_{3D}[\mathcal{V}](-\mathbf{k}, z_i) = \mathcal{F}_{3D}[\mathcal{V}](\mathbf{k}, z_i)^*, \tag{1}$$

and therefore query $D^2/2$ points through the neural representation (instead of $D^2$).

## C   Dataset Preparation

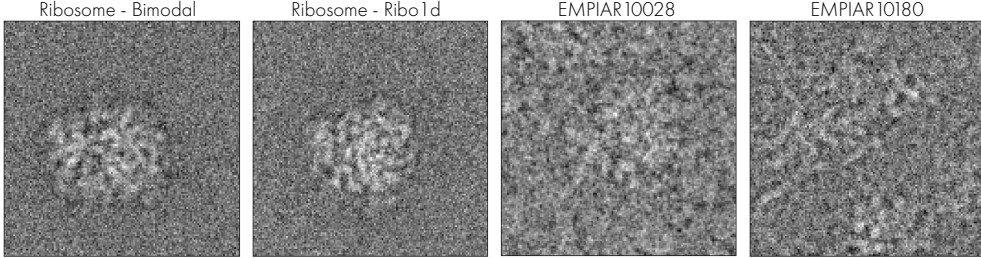

Figure S3: Sample images from the synthetic and experimental datasets.

Table S1: Summary of the parameters for synthetic and experimental datasets. $D$ is the resolution of the images, in pixels.

| Dataset | $D$ | $N$ | Å/pix. | Trans. | Classes |
|---|---|---|---|---|---|
| *Bimodal* | 128 | 50k-5M | 3.77 | ±8 pix. | 2 |
| *Ribo1d* | 128 | 100k | 3.77 | ±8 pix. | 2 / 3 / 10 |
| *EMPIAR-10028* | 128 | 105,247 | 3.77 | N/A | N/A |
| *EMPIAR-10180* | 128 | 139,722 | 4.25 | N/A | N/A |

*Synthetic 80S ribosome — "Bimodal"*

We use two reconstructed volumes of the *Plasmodium falciparum* 80S ribosome in the *rotated* and *unrotated* state as the ground truth volumes for the *"Bimodal"* dataset. Volumes were originally reconstructed by cryoDRGN analysis of EMPIAR-10028 [6, 7] and downloaded from zenodo [8] ($D = 256$, 1.88 Å/pix). Any voxels with density below a manually chosen isosurface threshold of 0.05 were zeroed to remove background density. Volumes were then downsampled to $D = 128$.

With these two ground truth volumes, we then prepare three synthetic datasets of varying sizes – *small* (50k images), *medium* (500k images) and *large* (5M images). Images were generated using a simulation of the image formation model on ground truth volumes. Each dataset contains a mix of projections with $90\%$ sampled from the *unrotated* volume as the major state and the rest from the *rotated* volume as the minor state. Rotations were sampled uniformly from SO(3) and translations were sampled uniformly from $[-8 \text{ pix.}, 8 \text{ pix.}]^2$. CTFs were drawn randomly with a log-normal distribution over the defocus range from EMPIAR-10028. We add Gaussian noise with variance $\sigma^2 = 10$ to the normalized projections (SNR=$-10$dB). Example images are shown in Figure S3.

*Synthetic 80S ribosome — "Ribo1d"*

For the *Ribo1d* dataset, we simulate a continuous transition of the 80S ribosome between the rotated and unrotated state using 10 volumes along this path as the ground truth volumes. Instead of linearly interpolating the two rotated and unrotated state volumes, which produces nonphysical artifacts, volumes were generated by interpolating in the latent space of the trained cryoDRGN model deposited to zenodo [8]. Specifically we use cryoDRGN's graph traversal algorithm to generate the interpolation path between the two end states. The resulting volumes were pre-processed similarly as described in the *bimodal* dataset.

With these ten volumes, we then follow the same image formation model as the *bimodal* dataset to create synthetic datasets of 100k images that contain either an equal mixture of the two end states (*"2 class"*), three volumes from the 1st, 5th, and 9th volume in the trajectory (*"3 class"*), or all 10 volumes (*"10 class"*).

*80S ribosome (EMPIAR-10028 [6])*

Images of the *Plasmodium falciparum* 80S ribosome were downloaded from EMPIAR-10028, and downsampled to $D = 128$ (3.77 Å/pix).

*Precatalytic spliceosome (EMPIAR-10180 [2])*

Images of the precatalytic spliceosome were downloaded from EMPIAR-10180, and downsampled to $D = 128$ (4.25 Å/pix). We train on the filtered set of 139,722 images available at [8]. Particle images are shifted by their published poses, since the particles in this dataset are significantly out of center [10].

## D  Additional Results

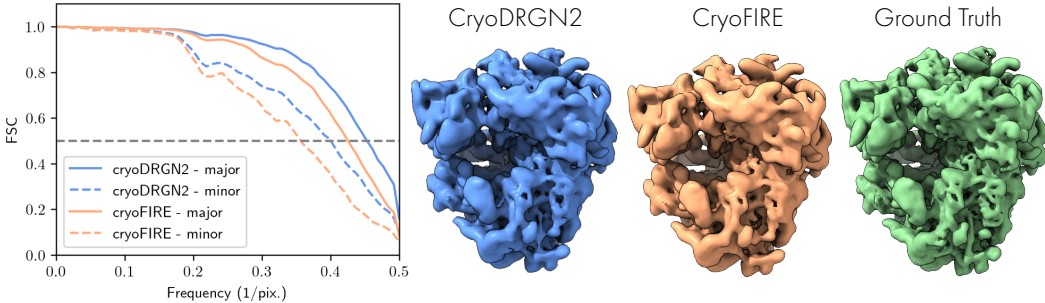

Figure S4: (Left) Fourier Shell Correlation (FSC) to the ground truth volume from the end of the training phase on the *large* (5M) ribosome *bimodal* dataset. (Right) Visualization of the reconstructed *major* volume.

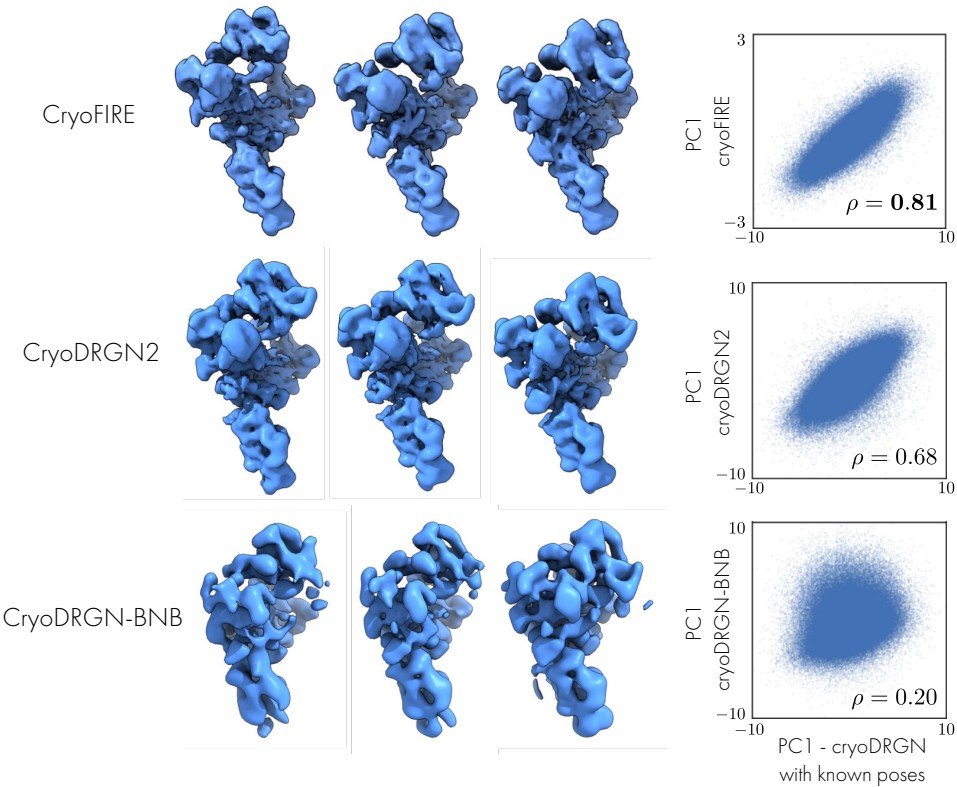

Figure S5: (Left) Volumes reconstructed by traversing along PC1 of the predicted $z_i$'s with cryoFIRE, cryoDRGN2 [10] and cryoDRGN-BNB [9]. (Right) Correlation between PC1 obtained with tested methods and PC1 obtained with cryoDRGN [7], where poses are given. Results from cryoDRGN2 and cryoDRGN-BNB are adapted from [10].

Figure S4 shows additional qualitative and quantitative results on the ribosome *bimodal* dataset. Figure S5 shows additional qualitative comparisons between cryoFIRE, cryoDRGN2 [10] and cryoDRGN-BNB [9] on the experimental precatalytic spliceosome dataset. See supplementary videos for a dynamic reconstruction obtained by traversing along PC1 with cryoFIRE.

Table S2: Comparison of cryoFIRE with state-of-the-art methods cryoSPARC [3] and cryoDRGN2 [10] on the *bimodal* dataset. We report the rotation accuracy (Rot. in degrees, ↓), the translation accuracy (Trans. in pixels, ↓), the confusion error (↓) and the resolution (Res. in pixels, ↓). CryoSPARC did not converge after 70h on the *large* dataset (5M images).

| *Dataset / Method* | Time | Rot. (Med/MSE) | Trans. (Med/MSE) | Confusion | Res. (major/minor) |
|---|---|---|---|---|---|
| *Small* (50k) | | | | | |
| cryoSPARC | **0:53h** (10 ep.) | 1.2 / 1.2 | 0.02 / 0.02 | 0.008 | **2.1 / 2.2** |
| cryoDRGN2 | 1:21h (20 ep.) | **0.8 / 0.8** | **0.007 / 0.01** | **0.00005** | 2.4 / 2.8 |
| **cryoFIRE** | 1:33h (70 ep.) | 2.3 / 2.6 | 0.09 / 0.1 | 0.0004 | 2.6 / 3.2 |
| *Medium* (500k) | | | | | |
| cryoSPARC | 9:27h (2 ep.) | 1.2 / 1.3 | 0.02 / 0.02 | 0.007 | **2.2 / 2.2** |
| cryoDRGN2 | 5:10h (2 ep.) | **0.8 / 0.9** | **0.007 / 0.01** | 0.002 | 2.5 / 3.0 |
| **cryoFIRE** | **1:28h** (7 ep.) | 2.7 / 2.9 | 0.1 / 0.2 | **0.0008** | 2.7 / 3.2 |
| *Large* (5M) | | | | | |
| cryoSPARC | > 70h | — | — | — | — |
| cryoDRGN2 | 21:37h (1 ep.) | **0.8 / 1.6** | **0.01** / 1.2 | 0.002 | **2.3 / 2.6** |
| **cryoFIRE** | **1:55h** (1 ep.) | 1.5 / 1.7 | 0.1 / **1.0** | **0.0002** | **2.3** / 2.7 |

Table S2 compares cryoFIRE with two state-of-the-art heterogeneous reconstruction methods: cryoSPARC [3] and cryoDRGN2 [10]. All these methods can process the data batch-wise, but cryoSPARC implements a fully non-amortized method where latent variables are inferred with an approximate Expectation-Maximization algorithm while cryoDRGN2 only amortizes the inference of the conformational state $z_i$. We run cryoSPARC v3.2.0 heterogeneous *ab initio* reconstruction with $K = 2$ classes using all default settings. The advantage of amortized approaches in terms of runtime is clearly visible on the *medium* dataset.

We run cryoFIRE on a published dataset of the 80S ribosome (EMPIAR-10028 [6]) for 300 epochs and show quantitative and qualitative results in Fig. S6. The dataset is filtered following [7]. The quantitative results validate the accuracy of the pose prediction (including translations) on a real dataset. See supplementary videos for a dynamic reconstruction.

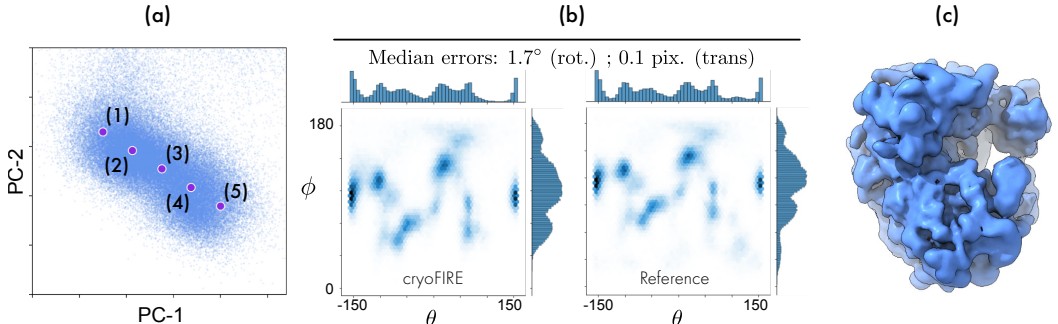

Figure S6: *Ab initio* reconstruction of an experimental dataset of the 80S ribosome (EMPIAR-10028 [6]). (a) Distribution of the two first principal components in the conformational landscape. (b) Distribution of the view directions obtained with cryoFIRE and published in the "reference" [6]. We indicate the median errors obtained on the predicted rotations and translations. (c) Reconstructed volume when sampling the point (3) in the conformational space (a). See supplementary videos for the reconstructed volumes (1) to (5).

## E   Discretization of the Image Formation Model

Although the image formation model described in Section 3 is continuous, the image collected on the detector is discrete. Assuming the electrostatic potential of the molecule is supported on a ball of radius $S/2$ ($S$ being the length of the size of the detector) and assuming the volume $\mathcal{V}(., z_i)$ is "smooth" (the coefficients of $\hat{\mathcal{V}}(., z_i)$ decay rapidly), we show here that our method reconstructs the volume $\mathcal{V}(., z_i)$ even though the discretization is not explicitly modeled in the decoder.

In real space, the (noise-free) image formation model can be modeled as

$$Y_i = T_{\mathbf{t}_i} * P_i * Q_i \tag{2}$$

If $s$ is the size of a pixel, the pixel located at $(x_{k,l}, y_{k,l}) \in \mathbb{R}^2$ receives an intensity

$$I_{k,l}^{(i)} = \int_{x_{k,l}-s/2}^{x_{k,l}+s/2} \int_{y_{k,l}-s/2}^{y_{k,l}+s/2} Y_i(x,y)dxdy = (A_{2\text{D}} * Y_i)(x_{k,l}, y_{k,l}) \tag{3}$$

where

$$A_{2\text{D}}(x,y) = \begin{cases} 1 & \text{if } |x| \leq \frac{s}{2} \text{ and } |y| \leq \frac{s}{2} \\ 0 & \text{otherwise.} \end{cases} \tag{4}$$

$\hat{I}^{(i)}$, the discrete Hartley transform of the image $I^{(i)}$ is defined by

$$\hat{I}_{m,n}^{(i)} = \sum_{k,l=1}^{D} \left( \cos\left(2\pi \frac{km}{D}\right) + \sin\left(2\pi \frac{ln}{D}\right) \right) I_{k,l}^{(i)}, \tag{5}$$

where $D$ is the number of pixels along each side of the detector. Since the locations of the pixels are

$$x_{k,l} = ks = k\frac{S}{D} \quad ; \quad y_{k,l} = ls = l\frac{S}{D}, \tag{6}$$

$$\hat{I}^{(i)}_{m,n} = \sum_{k,l=1}^{D} \left( \cos\left(2\pi x_{k,l}\frac{m}{S}\right) + \sin\left(2\pi y_{k,l}\frac{n}{S}\right)\right)(A_{\text{2D}} * Y_i)(x_{k,l}, y_{k,l}), \tag{7}$$

which can be re-written

$$\hat{I}^{(i)}_{m,n} = \mathcal{H}_{\text{2D}}(\mathcal{D} \odot A_{\text{2D}} * Y_i)\left(\frac{m}{S}, \frac{n}{S}\right), \tag{8}$$

where $\mathcal{D}$ corresponds to the sampling operation:

$$\mathcal{D}(x,y) = \sum_{k,l=1}^{D} \delta(x - ks)\delta(y - ls), \tag{9}$$

with $\delta$ the $\delta$-Dirac function. Finally, using the Fourier Slice Theorem we get

$$\hat{I}^{(i)}_{m,n} = \left[\mathcal{H}_{\text{2D}}(\mathcal{D}) * \mathcal{H}_{\text{2D}}(A_{\text{2D}}) \odot \hat{T}_{\mathbf{t}_i} \odot C_i \odot \mathcal{S}_i[\hat{\mathcal{V}}(.,z_i)]\right]\left(\frac{m}{S}, \frac{n}{S}\right) \tag{10}$$

$$= \left[\hat{T}_{\mathbf{t}_i} \odot C_i \odot \mathcal{S}_i[\mathcal{H}_{\text{3D}}(A^{(i)}_{\text{3D}}) \odot \hat{\mathcal{V}}(.,z_i)])\right]\left(\frac{m}{S}, \frac{n}{S}\right) \tag{11}$$

where

$$A^{(i)}_{\text{3D}}(x,y,z) = \begin{cases} 1 & \text{if } \|R_i \cdot [x,y,z]^T\|_\infty \leq \frac{s}{2} \\ 0 & \text{otherwise.} \end{cases} \tag{12}$$

$\mathcal{H}_{\text{2D}}(\mathcal{D})$ can be removed using the assumption that the signal is band-limited in real-space. Due to the spatial averaging in real space, the reconstructed volume in Hartley space is $\mathcal{H}_{\text{3D}}(A^{(i)}_{\text{3D}}) \odot \hat{\mathcal{V}}(.,z_i) \simeq \mathcal{H}_{\text{3D}}(A^{(0)}_{\text{3D}}) \odot \hat{\mathcal{V}}(.,z_i)$ if $|\hat{\mathcal{V}}(\mathbf{k}, z_i)|$ decays rapidly with $|\mathbf{k}|$, where

$$A^{(0)}_{\text{3D}}(x,y,z) = \begin{cases} 1 & \text{if } \|[x,y,z]\|_\infty \leq \frac{s}{2} \\ 0 & \text{otherwise.} \end{cases} \tag{13}$$

In real space, the continuous volume is "blurred" by a 3D window function of size $s$.

## F    Other External Softwares

Fourier Shell Correlations are computed using the software EMAN v2.91 [5]. Molecular graphics and analyses performed with UCSF ChimeraX [1], developed by the Resource for Biocomputing, Visualization, and Informatics at the University of California, San Francisco, with support from National Institutes of Health R01-GM129325 and the Office of Cyber Infrastructure and Computational Biology, National Institute of Allergy and Infectious Diseases.