# OpenReview forum: "Amortized Inference for Heterogeneous Reconstruction in Cryo-EM"
_NeurIPS.cc/2022/Conference — NeurIPS 2022 Accept_

### Official Review · Reviewer_7QvN · 2022-07-10

**Rating:** 7
**Confidence:** 4
**Soundness:** 4 excellent
**Presentation:** 3 good
**Contribution:** 4 excellent

**Summary:**

The paper proposes an ab initio heterogeneous reconstruction method for SPA cryoEM, named cryoFIRE. There are multiple advantages to the proposed approach and most of them come from the ability to jointly estimate both poses and conformations, allowing to skip computationally expensive step of pose search. The authors show the effectiveness of the method on multiple datasets (including one experimental) and compare the approach with SOTA of cryoEM NN-based heterogeneous reconstruction (cryoDRGN/2). The authors promise to provide an open-source implementation of cryoFire upon publication.

**Questions:**

1. The datasets used for the experiments have quite high SNR (especially synthetic ones), at least judging by the samples in the figure S3. Did you explore whether there is a limit of SNR at which the method does not work?
2. Even though Table 1 and Table S2 show that cryoFIRE is slightly bit less precise than cryoDRGN2 and cryoSPARC, the speedup in training and especially testing is more than worth the tradeoff. However, what do you think is the reason for decreased precision? What is the number of parameters in the models and have you experimented with changing number of parameters/layers?


**Limitations:**

The authors adequately addressed the limitations and potential negative societal impact of their work.

**Strengths And Weaknesses:**

Strengths:
1. The paper is written cleanly and I especially enjoyed the succinct formal description of cryoEM image formation.
2. The method is novel and presents the first instance of amortised inference for ab initio cryoEM heterogeneous reconstruction.
3. Joint estimation of poses and conformations brings one order of magnitude speedup, scaling much better with the ever-increasing amount of data produced by cryoEM.

Weaknesses:
1. As the authors themselves point, parameterisation calculated with cryoFIRE is not uniquely defined ("a rotation of the molecule can be equivalently represented by R_i or by a change of conformation state z_i"). The proposed workaround in the paper is to use training schedule and have "pose-only" phases at the start of training and then every N epochs. Additional work is required to be able to decouple rotation from conformation without training schedules.
2. cryoFIRE is directly compared with cryoDRGN and cryoDRGN2 (Table 1), as well as cryoSPARC (Table S2). I don't think that this represents software frameworks currently used by cryoEM groups.

---

> ### Author Response · Authors · 2022-08-02
> **Increased SNR and pose estimation**
>
> We thank the reviewer 7QvN for spending time on reviewing and providing insightful comments regarding the paper. We appreciate the reviewer acknowledging the clarity of the text, the novelty of the method and its impact on the field of cryo-EM. Below are our answers to the reviewer’s questions and some comments regarding how to decouple poses and conformations.
>
> [SNR in synthetic datasets] To exhibit more challenging noise conditions, we lowered the SNR from 0dB to -10dB and updated the results in Table 1 and Figure 3. CryoFIRE remains 10x faster than previous methods on the large dataset and reaches a similar resolution to that reached by cryoDRGN2. We did not explore an SNR limit where the method would fail, however we do test on two real cryo-EM datasets (spliceosome and ribosome). Future work could explore the performance of the method on even lower SNR synthetic or real images, especially of smaller protein complexes.
>
> [Decreased precision] We believe the decreased precision is a direct reflection of the fact that amortization maximizes a non-tight lower-bound of the likelihood, allowing for faster inference at the cost of accuracy (see [1]). Increasing the number of parameters in the encoder did not turn out to increase the accuracy of pose prediction. Future work could investigate a hybrid approach that would estimate poses with an encoder at the beginning of training and switch, at the end, to a local pose search initialized from encoder-estimated poses. This direction is suggested in the conclusion (L.334).
>
> [Decoupling poses and conformations] The pose only phase suggested in the paper does not, as the reviewer pointed out, completely decouple the pose from the latent conformational state. Future work in this direction is considered, like adding an explicit penalization on the mutual information between R and z. Although this coupling makes the conformational space difficult to interpret directly, it does not impact the quality of the volumetric reconstruction.
>
> [1] Donnat, Claire, et al. "Deep Generative Modeling for Volume Reconstruction in Cryo-Electron Microscopy." arXiv preprint arXiv:2201.02867 (2022)

---

### Official Review · Reviewer_hGi9 · 2022-07-10

**Rating:** 5
**Confidence:** 4
**Soundness:** 2 fair
**Presentation:** 3 good
**Contribution:** 2 fair

**Summary:**

The paper proposes a novel method to infer both pose and conformational states of protein structures from raw cryo-EM images in a way that is amortized over the size of the dataset. The authors compared the proposed method with cryoDRGN2 on several synthetic and experimental datasets. They demonstrated that their method requires less time compared to cryoDRGN2 by sacrificing little accuracy.


**Questions:**

1. How good is cryoFIRE in conformation estimation compared to other methods?
2. Why Rosembaum et al.’s method can not be applicable to a real dataset?
3. I would suggest taking a look at methods like SpatialVAE [1] and Harmony [2] that disentangles the rotation and translation from cryo-EM images. How are those methods related to yours? The setup of the discriminative model (Section 3.3) looks very similar.

[1] Bepler, Tristan, et al. "Explicitly disentangling image content from translation and rotation with spatial-VAE." Advances in Neural Information Processing Systems 32 (2019).

[2] Uddin, Mostofa Rafid, et al. "Harmony: A Generic Unsupervised Approach for Disentangling Semantic Content From Parameterized Transformations." Proceedings of the IEEE/CVF Conference on Computer Vision and Pattern Recognition. 2022.


**Limitations:**

The authors discussed the limitations and possible negative social impact of the work.


**Strengths And Weaknesses:**

The paper is well-written and overall easy to follow. The motivation and importance of the problem are well discussed. The paper provides a thorough overview of the literature. The proposed method is novel to the best of my knowledge.

However, there are several issues with the experiments, baselines, and results. The authors mentioned that Rosenbaum et. al.’s method was the only one to perform amortization for both pose and conformation estimation. In such a case, that particular method should have been kept as a baseline, not cryoDRGN2 since it amortizes for conformation estimation only. Even the comparison with cryoDRGN2 is not very impressive. It is quite usual that methods with an exhaustive search for pose would take more time with more accuracy. If the proposed method, cryoFIRE could achieve better or similar accuracy with less time then it would have been much more impressive. But cryoFIRE performs much worse than cryoDRGN2 even in large datasets.

No comparison has been showed for conformation estimation. Is cryoFIRE as good as cryoDRGN2 for conformation estimation as well? Can it recover some conformations that can not be recovered by cryoDRGN2 or 3DVA or E2GMM?

The paper mentions Rosembaum et al’s method only worked on the simulated dataset, but it did not discuss why it can not be applied to real datasets. What makes it inapplicable for real datasets but applicable for realistically simulated datasets? Or is it the case that the authors are the first to apply the method to the experimental dataset, this is clearly not a big contribution.

---

> ### Author Response · Authors · 2022-08-02
> **Baselines and related work**
>
> We thank the reviewer hGi9 for spending time on reviewing and providing insightful comments regarding the paper. We believe there is a misunderstanding in the relationship between our method and prior work, and we have clarified the reason why cryoDRGN2 is a relevant baseline to evaluate cryoFIRE and why this is not the case for Rosenbaum et al.’s work [1]. As the reviewer points out, although cryoDRGN2 reaches a better pose accuracy than cryoFIRE, we believe the practical difference is marginal since both methods achieve similar resolution in the final reconstruction, and that the decreased precision of the amortized inference approach is worth the order of magnitude training speedup
>
> [Baselines - Rosenbaum et al.’s work] We clarified a fundamental difference between Rosenbaum et al.’s work [1] and cryoFIRE in the related section (L.125): [1] relies on an atomic (RBF-based) representation of the molecule and needs to be initialized from a “base” backbone due to exponentially decaying gradients. Their method uses prior information about the atomic structure of the molecule, but is hard to optimize by gradient descent. The method was not demonstrated to work on real datasets. On the contrary, our method represents the volume with a neural network (coordinate-based representation), which does not embed prior structural information but can easily be optimized with gradient-descent-based techniques. In order to demonstrate the computational advantage of amortized inference, we compare cryoFIRE to cryoDRGN2 [4], which also uses a neural representation but infer poses with a pose search step.
>
> [Pose Accuracy] In order for the simulated experiments to be closer to a realistic setting, we lowered the SNR from 0dB to -10dB and updated the results in Table 1 and Figure 3. Although cryoDRGN2 reaches an accuracy 2 to 10 times higher than cryoFIRE on poses, the error on the translations remains below 0.2 pixel for cryoFIRE and the resolutions of the reconstructed volumes do not drop by the same factor compared to cryoDRGN2. In terms of resolution, which truly determines the quality of a reconstruction method, cryoFIRE reaches similar results as cryoDRGN2. CryoFIRE remains 10x faster than previous methods on the “large” dataset. Designing an hybrid method that would start by estimating the poses with an encoder and fine-tune them at the end with one step of pose search is an interesting direction for future work that could close the gap between cryoFIRE and cryoDRGN2 on the pose accuracy.
>
> [Conformation Accuracy] In Table 1, the “confusion” quantifies the accuracy of conformation estimation and shows that cryoFIRE and cryoDRGN2 give a similar accuracy. We did not show yet that cryoFIRE could recover conformations that would remain undetected by cryoDRGN2, 3DVA or E2GMM.
>
> [Disentangling semantic content] We added SpatialVAE [2] and Harmony [3] on L.118 of the related work and introduced them as demonstrations that VAE-based methods can accurately recover the latent parameters of transformations.
>
> [1] Rosenbaum, Dan, et al. "Inferring a continuous distribution of atom coordinates from cryo-EM images using VAEs." arXiv preprint arXiv:2106.14108 (2021)
> [2] Bepler, Tristan, et al. "Explicitly disentangling image content from translation and rotation with spatial-VAE." Advances in Neural Information Processing Systems 32 (2019)
> [3] Uddin, Mostofa Rafid, et al. "Harmony: A Generic Unsupervised Approach for Disentangling Semantic Content From Parameterized Transformations." Proceedings of the IEEE/CVF Conference on Computer Vision and Pattern Recognition. 2022
> [4] Zhong, Ellen D., et al. "CryoDRGN2: Ab initio neural reconstruction of 3D protein structures from real cryo-EM images." Proceedings of the IEEE/CVF International Conference on Computer Vision. 2021

---

> > ### Comment · Reviewer_hGi9 · 2022-08-07
> > **Response to author rebuttal**
> >
> > The authors sufficiently addressed my concerns. I think the method makes a good contribution to the cryo-EM community, however, the contribution of machine learning is still too specific. Thereby, I am increasing my rating.

---

### Official Review · Reviewer_ypJe · 2022-07-11

**Rating:** 6
**Confidence:** 4
**Soundness:** 3 good
**Presentation:** 3 good
**Contribution:** 3 good

**Summary:**

The paper describes a method, CryoFIRE, to recapitulate the 3D structure of a biomolecule from cryo-EM images using coordinate-based neural model that enables continuous representation. This is inspired by recent advancements of CryoDRGN/CryoDRGN2 for ab initio 3D molecular reconstruction. Different from existing methods, the authors propose to use an encoder to jointly estimate the particle poses and conformation state (a latent vector) from a set of 2D projections. The predicted poses and latent vector are then fed into a feedforward neural network to aggregate each particle image into an implicit representation of the scattering potential volume. The encoder and decoder are trained in an end-to-end fashion using L2 distance. This method achieves comparable results on both simulated and experimental datasets to SOTA. In addition, the paper shows that such autoencoder-based method can significantly speed up the reconstruction time compared to CryoDRGN2 when using a large number of 2D projections.

**Questions:**

Here are my detailed review comments.

1), One of the claimed contributions of this work is confusing, and some clarifications will be helpful. In specific, this work claims that “the first instance of amortized inference for ab initio heterogeneous reconstruction of experimental cryo-EM datasets.” Seems that CryoFIRE uses the same publicly available dataset as cryoDRGN2, and directly takes the results from cryoDRGN2.

2), The discussions around the differences and contributions over existing CryoAI is missing. Both methods use an encoder to capture the high-level poses of particle images and symmetrized loss.  In addition, CryoAI proposes to use two MLPs to represent the complex signals as phase and magnitude separately, while this work seems not adopt this strategy. What results in this different?

3), The rational and advantages of using the symmetrized loss during training in Sec 3.5 is not very instructive at its current state. A deeper understanding and intuition with some ablation studies will be helpful. In addition, how is the symmetrized loss generalizable? Would other training losses such as the statistical distances used in CryoDRGN benefit from this in-plane constrain?

4), While explicitly estimating the poses and amortized variations for parametrizing 3D structure is an interesting direction, recent work in deep learning such as CryoGAN [R1], for single particle cryo-EM has attempted to recover the 3D structure without explicitly recovering the poses using adversarial networks. It may be worth to discuss the relations and differences in the revision.

5), Consider explaining CryoDRGN2 in detail given that is the state-of-the-art technique and to which all the CryoFIRE results are compared.

6), What would runtime comparisons be for cryoDGRN, for an unsupervised heteregeneous reconstruction?

7), Consider adding more description of the neural network architecture that is the main contribution of this work (more than what is reported on page 5 and page S1 in the supplement).

[R1] Gupta et al., "CryoGAN: A New Reconstruction Paradigm for Single-Particle Cryo-EM Via Deep Adversarial Learning," in IEEE TCI, 2021.

Minor comments:

Figure 1 (c) is not very informative; it is hard to be linked to Figure 2.


**Limitations:**

The limitations of current method are missing. Considering single particle cryo-EM is an important but challenging problem, some discussions will be helpful (future work).

**Strengths And Weaknesses:**

Strengths:

The paper presents a novel approach for single particle cryo-EM that quantitatively and qualitatively works well. In addition, the paper is overall well written, and the method is well evaluated on experimental dataset.

Weaknesses:

Some claimed contributions and technical details might need to be clarified and discussed.

---

> ### Author Response · Authors · 2022-08-02
> **Contribution and related work**
>
> We thank the reviewer ypJe for spending time on reviewing and providing insightful comments and questions on our paper. We appreciate the reviewer acknowledging the novelty of the method and the quality of validating experiments, and we have provided clarifications to the questions below.
>
> [Contributions] We clarified in the introduction that cryoFIRE is able to amortize inference of the latent pose variables, unlike cryoDRGN2, which performs an explicit search over poses. We compare against the results of cryoDRGN2 on an experimental dataset to show that this approach can successfully reconstruct the same qualitative motions on real data.
>
> [cryoAI] Unlike cryoAI, our method performs heterogeneous reconstruction by feeding a latent vector z into the neural representation, which we have clarified in the related work section. We are representing the Hartley transform of the volume instead of its Fourier transform and we did not observe any improvement when using two MLPs instead of one. In cryoAI, both networks are SIRENs, i.e. contain sinusoidal activation functions in the hidden layers. With this kind of nonlinearity, the output can vary rapidly with the value of the input. However, we want the volume to smoothly depend on the latent variable z. We therefore chose to apply positional encoding only on the variable k and used ReLU activation functions, which led to significantly better heterogeneous reconstructions.
>
> [cryoGAN] A reference to and a comparison with CryoGAN were added in L.116. We emphasize the fact that pose estimation is not strictly necessary for the volume reconstruction and cryoGAN avoids inferring the poses via the use of a discriminative loss.
>
> [Future work] A direction for future work is suggested in the conclusion: investigating a hybrid approach that would estimate poses with an encoder at the beginning of training and switch, at the end, to a local pose search initialized from encoder-estimated poses.
>
> [Symmetric loss] An ablation study on the symmetric loss is done in cryoAI [2] and we now refer to this paper on L.220. In cryoDRGN2 [1], the authors tried to replace the pose search step with a variational encoder and called the method “PoseVAE”. As shown in the Supplement of [1] (Fig. S2), this method does not work on the “hand dataset”. In the Supplement of [2] (Fig. S6), the PoseVAE method is reproduced with and without the symmetric loss and it is shown that the symmetric loss is instrumental in enabling an accurate reconstruction. We now put more emphasis on the role of the symmetric loss on L.221.
>
> On question (6) about a runtime comparison, we are unsure what "unsupervised heterogeneous reconstruction" means in this context, perhaps a comparison to the runtime of cryoDRGN1 with known poses?
>
> [1] Zhong, Ellen D., et al. "CryoDRGN2: Ab initio neural reconstruction of 3D protein structures from real cryo-EM images." Proceedings of the IEEE/CVF International Conference on Computer Vision. 2021
> [2] Levy, Axel, et al. "Cryoai: Amortized inference of poses for ab initio reconstruction of 3d molecular volumes from real cryo-em images." arXiv preprint arXiv:2203.08138 (2022)

---

> > ### Comment · Reviewer_ypJe · 2022-08-09
> > **Response to authors**
> >
> > Thank the authors for the detailed replies! It resolves most of my concerns.  I have no further questions and believe that weak acceptance is a proper recommendation for the work.

---

### Official Review · Reviewer_fKm9 · 2022-07-11

**Rating:** 5
**Confidence:** 5
**Soundness:** 3 good
**Presentation:** 4 excellent
**Contribution:** 3 good

**Summary:**

This manuscript introduces a new method for the reconstruction of continuous variability in cryogenic electron-microscopy (cryo-EM) based on amortized inference and neural network autoencoders. While amortized inference and autoencoders have previously been used for different aspects of this problem, the proposed method provides a more comprehensive approach. Specifically, previous methods have relied on a consensus model for pose estimation or performed expensive pose estimation as part of an alternating refinement process where the main focus has been on estimating the latent states of the molecule. The proposed method performs the pose estimation at the same time as the latent state estimation, achieving significant speedups with respect to the state of the art. The method is validated on numerous synthetic datasets and one experimental dataset.


**Questions:**

– Please provide a formal definition for amortized inference early on (Section 1 or 2) for context.
– What does “disentanglement” mean in the context of cryoVAEGAN (line 108)?
– Another approach for joint estimation of pose and latent state (similar to 3DVA of cryoSPARC) is presented by the hypervolume model of Lederman et al. (2020). It would be good to compare this approach with the proposed model.
– What is calligraphic F in line 134?
– Section 3.1 describes an image formation model entirely in the continuous domain. Please discuss issues of discretization (since we are dealing with discrete data, after all). How is this taken into account?
– Where does D come from in line 156?
– How and why is the rotation “represented in the 6-dimensional space S² × S²”?
– “defoci” on line 196 should be “defocus” (plural is on “parameters”).
– What is meant by major and minor on line 224? (This is explained in the supplement, but needs to be briefly described in the main text.)
– What is meant by “images are masked with a circle of radius 32 pixels for pose search” on line 228?
– Similarly, what is meant by “pose search is done every 5 epochs”? Given that there is no search per se, does this mean that the inferred pose parameters are held fixed (i.e., not recomputed using the corresponding MLP) for 5 epochs in a row before the corresponding weights are updated?
– Why split the data into training and testing? Is this merely for validation purposes or part of the overall workflow? If it is part of the overall workflow, how is a user supposed to perform this split and how is the final reconstruction obtained?
– On line 243, the error is the “ratio of misclassified images” to what quantity?
– In Table 1, why is resolution not reported for the test sets?
– What is meant by “conditioning” on line 245?
– What is K on line 267?
– The Spearman correlation on line 271 is calculated between what variables?
– Why is only PCA of dimension 1 considered in Section 4.2.


**Limitations:**

As stated above, the high SNRs of the synthetic data, limited experimental validation, and lack of validation for translational pose estimation in the experimental data detract from the overall claims of the manuscript and should either be resolved or acknowledged in the text.

The authors discuss potential negative societal impact of their work (and the field in general) quite well.


**Strengths And Weaknesses:**

The main components of the proposed method have been proposed earlier: amortized inference for pose estimation has been introduced by Miolane et al. [15] and for latent state estimation by Zhong et al. [33]. Joint estimation of both pose estimation has been performed by Rosenbaum et al. [24], but only for synthetic data. These previous results are accurately described in the manuscript. As such, the original work in this manuscript consists of combining these existing components into a functioning whole, which is no easy feat. It also does this while achieving significantly reduced running times compared to the state of the art (albeit with a slight loss of accuracy).

The method is promising and the authors provide numerical results to support their claims. That being said, these numerical results do not completely validate the method as they currently stand. First, the synthetic data generated has a very high signal-to-noise ratio (SNR) by cryo-EM standards. This can be seen most clearly by comparing the sample projections in Figure S3, where two synthetic projection images are compared to an experimental projection image. The noise level of the former is much lower compared to the latter. In the text, the authors simply mention that the variance of the noise applied is one without providing a measure of the variance of the clean images. Judging by eye, I would guess that the SNR for the synthetic images is close to one, while realistic cryo-EM data typically has SNRs ranging from one hundredth to one tenth. It is important to note that despite these high SNRs, the proposed method performs worse than the state-of-the-art, with rotation errors on the order of 2× and translation errors typically on the order of 10×.

The results on the experimental data is also concerning. In the supplementary material, the authors mention that “particle images are shifted by their published poses, since the particles in this dataset are significantly out of center”. In other words, the pose estimation is not fully validated on experimental data (specifically, while the rotational component of pose estimation is validated, the translational component is not). While it may well be the case that the particles in this dataset are significantly out of center, this is a common occurrence in many cryo-EM datasets, and it is something that a robust method should be able to handle. Given the fact that an important contribution of the proposed method is its validation on experimental data, this discrepancy significantly detracts from that contribution. Furthermore, the fact that only one experimental dataset is tested also reduces the scope of validity (compared to the cryoDRGN and Relion multi-body refinement papers, which present results on three or more experimental datasets).

The overall presentation of the work is clear, with most concepts being well defined. Given the importance put on amortized inference, it would be good to specify early on what is meant in a formal manner to reduce any possible confusion about the term. In a similar vein, a definition of the Hartley transform (and its relative advantages over the Fourier transform) would be a good addition.o

As stated above, the significance of this paper is the joint amortized inference of pose and latent space applied to experimental data. As such, it provides an important methodological milestone for the processing of cryo-EM data. However, the numerical results do not completely back up these claims. Additionally, the authors also do not make clear what it is about their setup that makes the method successful. In other words, what has been missing from previous attempts at joint inference (such as the work of Rosenbaum et al. [24]) that is addressed by the current work?

---

> ### Author Response · Authors · 2022-08-02
> **Increased SNR and more real datasets**
>
> We thank reviewer fKm9 for spending time on reviewing and providing detailed comments that have helped us improve the paper’s quality. We appreciate the reviewer highlighting the challenging nature of the problem as well as the speed cryoFIRE achieves. Following the reviewer’s comments on more validation of the approach, we have repeated our synthetic experiments at a more realistic noise level and included results on an additional experimental dataset.
>
> [SNR in synthetic datasets] We lowered the SNR from 0dB to -10dB and repeated the experiments for Table 1 and Fig. 3, leading to the same conclusions, namely that cryoFIRE remains 10x faster than previous methods on the large dataset. We have noted that cryoFIRE has lower accuracy on pose estimation than cryoDRGN2, however, we believe that the difference is marginal as the final reconstruction reaches a similar resolution and was achieved with an order of magnitude speedup. We have added this discussion in the text and clarified the definition of the SNR (L.241).
>
> [Additional real datasets] We used the real spliceosome dataset for evaluation as it is the benchmark used in the cryoDRGN2 paper. We re-centered the images as this particular dataset is odd in that many of the particles are located very far from the image center. This bias can be easily corrected in refinement, and without this re-centering, the bias even leads to failure of cryoSPARC homogeneous reconstruction (see Table S5, S6 in the cryoDRGN2 paper). In order to validate that cryoFIRE accurately estimates translations on a real dataset, we included results for ab initio heterogeneous reconstruction of the 80S ribosome in the Supplement.
>
> [Key contribution] In [5], the reconstructed potential is a mixture of Radial Basis Functions based on an atomic model that can only be optimized if initialized close to the true structure (due to exponentially decaying gradients); the method was not proven to work on real datasets. Our method represents the volume with a neural network, which does not embed prior structural information but can easily be optimized with gradient-descent-based techniques.
>
> [Hyper-molecules] A reference to [1] was added in the related works as an example of method using MCMC for ab initio heterogeneous reconstruction and introducing a new mathematical framework for representing deformable molecules.
>
> [Discretization] A section was added in the Supplement to show that not taking into account discretization in the forward model does not impact the reconstruction for a molecule that has rapidly decaying coefficients both in real (compared with the size S of the detector) and Fourier (compared with 1/S) spaces.
>
> [S2xS2 representation] Rotations are represented in S2xS2 by normalizing two vectors of R3. This representation was shown to give better results for estimating the latent poses in [2]. Theoretical reasons for this can be found in [3]. This is now mentioned on L.195.
>
> [PCA] We only consider the first PC as it embeds most of the variance in the conformational space and makes both visualization and clustering easier. This method is imperfect as it does not well quantify the “sensitivity” (different conformations are associated to z’s that are “far” from each other) and the “specificity” (identical conformations are associated to z’s that are “close” from each other) of cryoFIRE. However, there does not seem to be a consensus in the cryo-EM community regarding how to quantitatively evaluate a reconstructed conformational landscape (especially in the case of a continuous movement).
>
> [Test set] The purpose of the test set is to show that the encoder does not only memorize the training set but can accurately infer the latent variables associated with images it has never seen. We do not report the resolution at test time since only the encoder is evaluated. This is clarified on L.251.
>
> [Terminology and notations] Clarification has been provided regarding important terminology: “amortized inference” (at the first occurrence, on L.47), “disentanglement” (L.114), Hartley transform (L.160), “masking” (L.243), “major/minor” (L.239), “pose search” (L.245), the “ratio of misclassified images” (L.260) and “conditioning” (L.261). The following notations were clarified: calligraphic F (set of functions from R3 to R, L.146), D (number of pixels, L. 169), K (number of conformations, L.279). The Spearman correlation mentioned on L.288 is calculated between the index of the ground truth conformation and the value of PC1.
>
> [1] Lederman, Roy R., et al, "Hyper-molecules" Inverse Problems 36.4 (2020)
> [2] Nashed, Youssef SG, et al. "Cryoposenet" CVPR, 2021
> [3] Zhou, Yi, et al, CVPR 2019
> [4] Levy, Axel, et al, arXiv preprint arXiv:2203.08138 (2022)
> [5] Rosenbaum, Dan, et al, arXiv preprint arXiv:2106.14108 (2021)

---

### Author Response · Authors · 2022-08-02
**Summary**

We thank the reviewers for their thoughtful and valuable feedback on the paper. We introduce cryoFIRE, a new method for ab initio heterogeneous reconstruction in cryo-EM that amortizes the inference of poses and conformational states over the size of the dataset. We appreciate the reviewers highlighting the novelty of the method and the speed cryoFIRE can achieve compared with the state-of-the-art.

We agree with the reviewers’ suggestion to use more realistic synthetic datasets and to validate cryoFIRE on more experimental datasets. Following their comments, we simulated noisier datasets and updated the results in Table 1 and Figure 3. We added qualitative and quantitative results obtained on an experimental dataset of the 80S ribosome in the Supplement. We clarified important terminology used throughout the paper, and clarified the contribution of this method with the related work.

---

### Meta-Review · Area_Chair_LqVF · 2022-09-02

**Recommendation:** Accept
**Confidence:** Certain

**Metareview:**

The paper studies the problem of simultaneously estimating poses and conformation of a biomolecule, cryo-EM images. It presents a pipeline which integrates reconstruction and conformation estimation, leading to significant time savings compared to methods that alternate between accurately estimating poses and conformations. Conventional pose estimation is expensive because it involves searching over the space of rigid body motions, by repeatedly rendering images from from various viewpoints. The paper proposes an autoencoder-like structure, which generates conformation and pose estimates for each image, which are used by a decoder to produce an estimate of the conformation.

Reviewers generally evaluated the paper positively, noting that it achieves an order of magnitude speedup compared to a conventional baseline. Reviewers note that although elements of the paper have previously appeared (the use of autoencoder-like architectures for cryo-EM reconstruction, the use of amortized inference over conformations and poses), the combination employed here is novel. Reviewers generally appreciated the paper’s experimental results, while raising some concerns about the SNR and baselines for comparisons. Finally, reviewers noted that the paper provided a clear exposition of both cryo-EM and the proposed techniques. Overall, the paper exhibits a well-chosen combination of learning techniques, which lead to performance improvements for a problem of significant scientific interest.

**Award:**

No

---

### Decision · Program_Chairs · 2022-09-14

Accept